# Ischemic Preconditioning with High and Low Pressure Enhances Maximum Strength and Modulates Heart Rate Variability

**DOI:** 10.3390/ijerph19137655

**Published:** 2022-06-23

**Authors:** Luiz Guilherme Telles, François Billaut, Aline de Souza Ribeiro, Christian Geórgea Junqueira, Luís Leitão, Ana Cristina Barreto, Patricia Panza, Jeferson Macedo Vianna, Jefferson da Silva Novaes

**Affiliations:** 1Physical Education and Sports Department, Federal University of Rio de Janeiro, Rio de Janeiro 21941-901, Brazil; guilhermetellesfoa@hotmail.com (L.G.T.); jeffsnovaes@gmail.com (J.d.S.N.); 2Estácio de Sá University (UNESA), Rio de Janeiro 20261-063, Brazil; 3Department of Kinesiology, Laval University, Quebec, QC G1V 0A6, Canada; francois.billaut@kin.ulaval.ca; 4Physical Education and Sports Department, Federal University of Juiz de Fora, São Pedro 36036-900, Brazil; alinevalencaedfisica@gmail.com (A.d.S.R.); paty_panza@yahoo.com.br (P.P.); jeferson.vianna@ufjf.edu.br (J.M.V.); 5Volta Redonda University Center, Rio de Janeiro 27240-560, Brazil; christian.junqueira@hotmail.com; 6Sciences and Technology Department, Superior School of Education of Polytechnic Institute of Setubal, 2910-761 Setúbal, Portugal; 7Life Quality Research Centre, 2040-413 Rio Maior, Portugal; 8Celso Lisboa University Center, Rio de Janeiro 20950-092, Brazil; educacaofisica@celsolisboa.edu.br

**Keywords:** ischemic preconditioning, maximum strength, resistance exercises, heart rate variability

## Abstract

Background: The application of ischemic preconditioning (IPC) to resistance exercise has attracted some attention, owing to increases in muscle performance. However, there is still no consensus on the optimal occlusion pressure for this procedure. This study compared the acute effects of IPC with high and low pressure of occlusion on upper and lower limb maximal strength and heart rate variability in recreationally trained individuals. Methods: Sixteen recreationally trained men (25.3 ± 1.7 years; 78.4 ± 6.2 kg; 176.9 ± 5.4 cm; 25.1 ± 1.5 m^2^ kg^−1^) were thoroughly familiarized with one repetition maximum (1 RM) testing in the following exercises: bench press (BP), front latissimus pull-down (FLPD), and shoulder press (SP) for upper limbs, and leg press 45º (LP45), hack machine (HM), and Smith Squat (SS) for lower limbs. The 1 RM exercises were then randomly performed on three separate days: after a high pressure (220 mmHg, IPC_high_) and a low pressure (20 mmHg, IPC_low_) IPC protocol and after no intervention (control, CON). Heart rate variability was also measured at rest, during and after the entire IPC protocol, and after the exercises. Results: Maximal strength was significantly (*p* < 0.05) higher in both IPC_high_ and IPC_low_ compared with CON in all upper- and lower-limb exercises. There was no difference between the two experimental conditions. No significant differences were found in the comparison across the different experimental conditions for LF_nu_, HF_nu_, LF/HF ratio, and RMSSD_ms_. Conclusions: IPC performed with both high and low pressures influenced heart rate variability, which may partly explain the maximal strength enhancement.

## 1. Introduction

The efficacy of ischemic preconditioning (IPC) has been investigated substantially in the health context in the last 34 years, and it has been robustly demonstrated that this maneuver induces protection of the heart such as a reduction in myocardial infarction area [1] and of the skeletal muscle such as a reduction in ischemic necrosis and lowering of energy metabolism during sustained ischemia [1]. In the context of physical exercise and performance, studies have mainly investigated the impact of IPC on endurance and power-oriented activities [2,3]. More recently, the application of IPC to resistance exercise has attracted some attention, demonstrating very interesting and relevant benefits on muscle performance and adaptations [4,5,6].

The improvement in muscle performance after application of IPC has been demonstrated for isometric [7,8], dynamic [9,10], and isokinetic strength [11]. Recently, Carvalho et al. [12] showed that IPC can further enhance the chronic increase in maximum strength after 6 weeks of intervention combined with resistance exercise. However, surprisingly, the acute effect of IPC on maximum strength has not yet been verified. According to the *American College of Sports Medicine* [13], maximum strength is defined as the highest resistance of a muscle or muscle group that can be moved along the range of motion of the joint, in a controlled manner and with the correct posture. Kilduff et al. [14] has suggested that there is an additional window on the day of competition during which performance can be enhanced with various acute strategies. Among them, IPC has been proposed to be such an efficient strategy. In fact, Winwood et al. [15] recommended the use of IPC for Strongman athletes on the day of the competition to improve maximum strength and muscle power. Recently, Telles et al. [16] demonstrated an increase in the number of repetitions at 80% 1 RM using IPC as a warm-up strategy for strength exercises in trained men.

In addition to the lack of data on maximal strength capacities in trained individuals, there is also a lack of consensus in the literature about the vascular occlusion pressures when using IPC. The pressures typically range from 10 to 300 mmHg [8,17], and it appears that both partial (low pressure) and total (high pressure) vascular occlusion can positively affect muscle performance. Marocolo and colleagues compared the effects of IPC with 220 mmHg to a SHAM procedure using 20 mmHg on muscle resistance during resistance exercise with the lower limbs [9] and upper limbs [10] and reported that IPC increased the number of maximum repetitions in both occlusion pressures. Similarly, De Souza et al. [5] reported improved muscle endurance at both high (220 mmHg) and low pressure (20 mmHg) when compared to CON. However, Paradis-Deschênes et al. [11] observed strength benefits during knee extensions with a high pressure (200 mmHg) only. Da Silva Novaes et al. [4] demonstrated that only 220 mmHg IPC procedure increased the total volume (i.e., the number of repetitions) within a resistance exercise session.

In the sport sciences, the low occlusion pressures (ex. 20 mmHg) have especially been used as SHAM treatments [4,16] to demonstrate the potency of ischemic episodes. However, some authors [5,9,10] have suggested that there may be a potentially beneficial psycho-physiological effect associated with low pressures. In fact, a pressure as low as 10% of arterial occlusion pressure in the right arm (which equates to 15 mmHg) has been reported to induce a 30% significant reduction in blood flow, and blood flow further decreased by almost 50% when arterial occlusion pressure was increased to 30 mmHg only [18].

In addition, some authors [17,19] have suggested that a possible neural stimulus acting through types III and IV afferent fibers may affect the neural drive, thereby increasing muscle strength. Along this line of reasoning, IPC has been shown to modulate the autonomic nervous system [20,21]. Heart rate variability (HRV) has been used to indirectly and non-invasively estimate sympathovagal balance after application of IPC [20,21,22]. Lopes et al. [21] demonstrated that IPC can accelerate recovery of HRV after high-intensity exercise. Gardner et al. [20] showed that 2 weeks of IPC application positively changed the sympathovagal balance in healthy individuals. In this context, the use of HRV during the application of IPC at different pressures could be useful to elucidate part of the mechanisms involved in the ergogenicity of low-pressure IPC.

Therefore, our goal was to evaluate the acute effects of ischemic preconditioning with high (IPC_high_) and low pressure of occlusion (IPC_low_) on upper and lower limbs maximum strength and heart rate variability in recreationally trained individuals. We hypothesized that both occlusion pressures would have positive effects on heart rate variability and maximum strength.

## 2. Materials and Methods

### 2.1. Experimental Design

The present study included 5 visits to the laboratory, separated by 3–7 days, always at the same time of day to avoid the circadian influence (Figure 1). In the first visit, all participants signed the informed consent form and fulfilled the Physical Activity Readiness Questionnaire/PAR-Q. Anthropometric data collection and a familiarization of the 1 repetition maximum (1 RM) testing procedures were also performed at that visit. During the second visit, 1 RM retest was performed to access the load for the three protocols. In the third, fourth, and fifth visits, all participants were randomly assigned to three experimental protocols: (a) high pressure IPC (220 mmHg) (IPC_high_) + 1 RM testing; (b) low pressure IPC protocol (20 mmHg) (IPC_low_) + 1 RM testing; (c) 1 RM testing CON (control protocol).

### 2.2. Sample and Ethical Procedures

The study included 16 men between 18 and 35 years of age, normotensive, physically active, and with at least one year of experience in resistance exercise (Table 1). Subjects that responded positively to any of the items *Physical Activity Readiness Questionnaire/PAR-Q* [23], missed one or more visits of the collection procedures in the laboratory, presented some type of osteoarticular injury in the upper or lower limbs, had been taking some medication or supplements in the last 3 months before the collection, and smokers, were excluded from the study. After being explained the risks and benefits of the research, the subjects signed the informed consent form prepared according to the Helsinki declaration and approved for human experiments by the Volta Redonda University Center’s local Ethics and Research Committee under protocol number 2,699,294. During the study period, participants were instructed to restrain from exercise as well as avoid caffeine, nutritional supplements, and alcohol intake 48 h before, during, and after the entire study, sleep for a minimum of six hours on the night before the test session, and not perform the valsava maneuver during the execution of the exercises. The average time between visits to the laboratory was 3.5 ± 0.6 days.

### 2.3. Procedures

#### 2.3.1. Ischemic Preconditioning Protocols

The IPC_high_ protocol session consisted of 4 cycles of 5 min of occlusion at 220 mmHg pressure using a 57 × 9 cm pneumatic tourniquet applied around the subaxillary region of the upper arm (komprimeterRiester^®^, Jungingen, Germany) with 5 min alternation of reperfusion at 0 mmHg. This resulted in a total intervention of 40 min. The pressure used and the breadth of the cuff are in accordance with previous studies [4,16]. In order to verify that the individuals had the blood flow obstructed during the intervention, the radial pulse was manually checked by digital palpation.

The IPC_low_ protocol session consisted of 4 cycles of 5 min of occlusion at 20 mmHg pressure, as proposed in previous studies [4,16], alternating with 5 min at 0 mmHg for a total of 40 min. The subjects remained seated during the intervention, lasting 40 min [4,16].

#### 2.3.2. Anthropometric Evaluation

Height and body mass were measured with a 0.5 cm precision stadiometer and a 0.1 kg precision Filizola^®^ scale, respectively. All measurements were performed following the recommendations of ACSM [13]. These measurements were subsequently equated to obtain the body mass index (BMI) in kg m^−2^.

#### 2.3.3. One Repetition Maximum (1 RM) Testing

The intervention’s load prescription was evaluated through the 1 RM test [13]. The evaluations were performed on the days of the laboratory visits as described in Figure 1. The flowing exercises were performed in that same sequence every time, with 10 min of rest in-between: bench press (BP), front latissimus pull-down (FLPD), and shoulder press (SP) for upper limbs, and leg press 45º (LP45), hack machine (HM), and Smith Squat (SS) for lower limbs. Individuals started warming-up with two sets of 5–10 repetitions at 40–60% of the maximum perception of the individual’s strength, separated by 1 min. After a 1 min interval, they performed a third set of 3–5 repetitions at 60–80% of the maximum perceived strength. After another minute of rest, the strength testing evaluation was initiated. Five trials were performed for every exercise, adjusting the load before every trial to obtain a precise evaluation of the 1 RM. The recovery time between the attempts was standardized in 5 min. The test was interrupted when the individual could not execute the movement correctly, being considered that repetition of the maximum load with the execution completed.

The following strategies were adopted to reduce the margin of error in data collection procedures: (a) standardized instructions given before the tests, so that each tested subject was aware of the routine involved in data collection, (b) the tested subject was instructed on the proper technique for performing the exercise, (c) all participants received standardized verbal encouragement during the tests, and (d) all tests were performed at the same time of day for each session. The highest load achieved between the two days was considered the 1 RM [4].

#### 2.3.4. Heart Rate Variability (HRV)

A Polar Electro Oy (Kempele, Finland) 800CX monitor was used 10 min before IPC maneuvers, during IPC maneuvers (every 5 min up to 40 min), and after each experimental protocol and always before performing the 1 RM tests, with the volunteers in the sitting position during the rest. During recordings, participants were asked to keep their eyes open, breathe calmly, and avoid movement. The cardiofrequencemeter had a sampling frequency of 1000 Hz, fixed by an elastic belt at the height of the sternum (1/3 lower) and with simultaneous transmission to the clock fixed on the left arm handle, where the register was stored. Later, through the infrared sensor serial port interface the data was transported and stored in the Polar Precision Performance program on an Acer^®^ branded computer. This data was exported and analyzed in Kubios HRV Analysis Program 2.0 software (version 2.2, Kuopio, Finland). After noise removal through visual inspection of the iRR distribution (ms), the most stable period over 5 min was selected [22] and time-domain and frequency-domain were extracted. For time-domain analysis, the square root of the sum of the differences between the R-R intervals divided by the number of R-R intervals was calculated, and for frequency-domain spectral analysis LF (low frequency), HF (high frequency), and LF/HF ratio were calculated.

### 2.4. Statistical Analysis

The results are presented in mean ± standard deviation values. The normality was verified by the Shapiro–Wilk test and the homoscedasticity was confirmed by the Levene test. One-way ANOVA with repeated measurements was used to test interactions and compare the means of the 1 RM tests. Significant differences were identified by Tukey’s post-hoc test. The two-way ANOVA analysis of variance for repeated measurements was performed to determine the differences in experimental protocols for dependent variables (RMSSD, HF_nu_, LF_nu_, LF/HF, SDNN, pNN50, and HF). To determine the specific differences, Tukey’s post-hoc test was performed. The Effect Size (ES) (Cohen’s d) estimates were calculated using the standardized mean difference to determine the magnitude of the treatment effects. The magnitude of each ES was interpreted using the scale proposed by Rhea [24]. The changes in (Δ%) were calculated for the variables (RMSSD, HF_nu_, LF_nu_, LF/HF, SDNN, pNN50, and HF) comparing the baseline moment with the moments after the exercises. All the analyses were performed in the SPSS software (SPSS Inc., V.21, Chicago, IL, USA) and considered an alpha value of 5% (*p* < 0.05). The sample size calculation [25] for the study was performed with G*Power (ver. 3.1.9.7; Heinrich-Heine-Universität Düsseldorf, Düsseldorf, Germany) with an N of 16 individuals for a power of 0.8, α = 0.05, correlation coefficient of 0.5, nonsphericity correction of 1, and an effect size of 0.32.

## 3. Results

High intra-class coefficients of correlation (ICCs) were found for the 1 RM test–retest for all exercises: BP (0.954), LP (0.976), LPD (0.964), HM (0.948), SP (0.990), and SS (0.988). All variables tested demonstrated a normal distribution (*p* > 0.05). The ES, *p*-values, and percentage changes (Δ%) for experimental conditions for LF_nu_, HF_nu,_ LF/HF ratio, RMSSD_ms_ SDNN_ms_, pNN50 (%), and heart rate for each condition and time point are presented in Table 2.

The maximal strength was significantly (*p* < 0.05) higher in both IPC_high_ and IPC_low_ compared with CON for BP, LPD, and SP (Figure 2). Significant protocol x treatment for BP (F_(15,30)_ = 63.2; *p* = 0.0001; eta^2^ = 0.954) displayed increases from CON with IPC_high_ and IPC_low_ (Figure 2). Significant protocol x treatment for LPD (F_(15,30)_ = 27.2; *p* = 0.0001; eta^2^ = 0.884) displayed increases from CON with HP and LP (Figure 2). Significant protocol x treatment for SP (F_(15,30)_ = 48.8; p= 0.0001; eta^2^ = 0.933) displayed increases from CON with IPC_high_ and IPC_low_ (Figure 2).

The maximal strength was significantly (*p* < 0.05) higher in IPC_high_ and IPC_low_ compared with CON for LP45, HM, and SS (Figure 3). Significant protocol × treatment for LP45 (F_(15,30)_ = 40.2; *p* = 0.0001; eta^2^ = 0.924) displayed increases from CON with IPC_high_ and IPC_low_ (Figure 3). Significant protocol × treatment for HM (F_(15,30)_ = 35; *p* = 0.0001; eta^2^ = 0.894) displayed increases from CON with IPC_high_ and IPC_low_ (Figure 3). Significant protocol x treatment for SS (F_(15,30)_ = 164; *p* = 0.0001; eta^2^ = 0.976) displayed increases from CON with IPC_high_ and IPC_low_ (Figure 3).

No significant differences were found in the comparison across the different experimental conditions for LF_nu_, HF_nu_, LF/HF ratio (Figure 4), RMSSD_ms_., SDNN_ms_., pNN50 (%), and heart rate (Figure 5). However, significant intra-protocol differences were identified. LF_nu_ increased (*p* < 0.05) from baseline at 10, 15, 25, 30, 35, 40 min and 5-min-post for IPC_high_ (Figure 4) and from baseline at 5, 10, 20, 25, 30, 40 min and 5-min-post for IPC_low_ (Figure 4). HF_nu_ decreased (*p* < 0.05) from baseline at 10, 15, 20, 25, 30, 35, 40 min and 5-min-post for IPC_high_ (Figure 4) and from baseline at 5, 10, 25, 30, and 40 min for IPC_low_ (Figure 4). LF/HF ratio increased (*p* < 0.05) from baseline at 5, 10, 20, 25, 30, and 40 min for IPC_low_ only (Figure 4).

Furthermore, RMSSD_ms_ decreased (*p* < 0.05) from baseline at 5, 10, 15, 20, 25, 30, 35, 40 min and 5-min-post for IPC_low_ only (Figure 5).

## 4. Discussion

The objective of this study was to evaluate the acute effects of ischemic preconditioning with high (IPC_high_) and low pressure of occlusion (IPC_low_) on upper and lower limb maximum strength and heart rate variability in recreationally trained individuals. The main findings were: (1) IPC with both pressures equally increased maximal strength for all exercises compared to CON testing; (2) IPC with both pressures induced higher activity of the low frequency (LF_nu_) and lower activity of the high frequency (HF_nu_) during the maneuver; and (3) only IPC with low pressure increased the sympathovagal balance (LF/HF ms2) and reduced the RMSSD-ms vs. resting values during the maneuver. To our best knowledge, this study is the first to investigate the acute effects of IPC on maximum strength development in six multi-articular, upper and lower limb resistance exercises.

IPC at high pressures (i.e., >200 mmHg) has consistently been shown to increase isometric [7,8] and dynamic strength [4,5,10], as well as the number of repetitions in exercises involving both lower [4,5,9] and upper limbs [4,10,16]. Thus, these transient vascular occlusions have been proposed as a potential warm-up strategy to enhance strength in trained men [16]. Although high pressures induce local hypoxia, benefiting from the hypoxia- and reperfusion-related complex cascade events, they do not always promote higher muscle performance as compared with a placebo application with low pressure. For example, both IPC and placebo interventions were reported to enhance acute muscle endurance during knee extensions [9]. Such benefits were also reported after 4–5 days of daily IPC and placebo application on maximal load during a 12-RM test of the elbow flexors [10] and endurance during knee extensions [5]. Our current results add to the literature by showing that IPC with high and low pressures may equally increase acute maximal strength capacity.

Marocolo et al. [9,10] and De Souza et al. [5] have suggested that there may actually be a psycho-physiological effect of the maneuver instead of only a placebo effect, which could explain the increased performance with both low and high occlusion pressures. However, unfortunately, they did not analyze any physiological nor psychological responses to IPC so their conclusions remain speculative. The current study is therefore the first to evaluate the heart rate variability during and after the maneuvers, in order to ascertain the presence of a possible neurophysiological effect that could explain the ergogenic effect of IPC_low_ observed here and elsewhere. Our findings showed that both IPC at 220 and 20 mmHg induced a predominance of the low frequency over the high frequency during the maneuvers. However, only the IPC_low_ increased the sympathovagal balance (LF/HF ms2) and reduced the RMSSD-ms.

Heart rate variability has been used to indirectly and non-invasively evaluate parasympathetic and sympathetic activity after application of IPC [20,21,22]. The changes observed during IPC_low_ may be related to a spontaneous inhibition of afferent fibers of type III and IV caused by the acute release of opioids, a specific group of autacoids, after the application of the IPC [26]. According to Cruz et al. [26], IPC can alter the receptors’ activation threshold, desensitizing the groups III and IV afferent fibers. Such phenomenon can maintain or even increase the neural drive and the number of motor units recruited; thus, increasing the strength production. Some studies have examined the autonomic nervous activity after IPC_high_ maneuvers [19,20,21]. Lopes et al. [21] investigated the effect of IPC_high_ on autonomic cardiac recovery after repeated high-intensity sprints. They demonstrated an increase in HR recovery within 60 s, which indicates a higher activity of parasympathetic nervous system (PNS) and a reduction in SNS activity. On the other hand, Incognito et al. [19] reported no impact of IPC_high_ on SNS activity up to 3 min after a handgrip isometric strength test. Enko et al. [27] (14) showed an increase in PNS activity and a reduction in noradrenaline after 30 min of IPC_high_ applied to the arms. Recently, Gardner et al. [20] demonstrated that 2 weeks of IPC_high_ (220 mmHg) changed the sympathovagal balance in healthy individuals. Importantly, all of the above studies used a high pressure and, to our best knowledge, there is no report of autonomous nervous system activity during and after a low-pressure maneuver. The pressure of 20 mmHg used in the current investigation did induce changes in the heart rate variability. This is in line with the observation that a pressure as low as 15 mmHg can cause significant physiological changes [18]. Therefore, the “placebo” effect that some studies reported [5,9,10], may represent a true physiological effect in some people. In fact, Mouser and colleagues [28] have suggested that the restriction pressure should be individualized since a given absolute pressure may induce completely different restriction values among different individuals. As a future direction in this field, it may be relevant to identify low- vs. high-pressure “responders” to determine the individual IPC limit of effectiveness and implement more comfortable, but still efficient, protocols for varied people.

Main methodological limitations of the study and perspectives include the following: (a) the short interval between the IPC maneuver and the maximum strength test may have interfered with the effects of IPC_high_, as reported in previous studies [28,29], and could have reduced the amplitude of change compared with IPC_low_. Thus, future research should elucidate the impact of such a methodological parameter in the efficacy of IPC in increasing maximum strength; (b) in the present study, the autonomic nervous system was not directly evaluated limiting our interpretation of the data; and (c) HRV is an indirect measure of cardiac vagal control, as it relies on heart rate measurements rather than direct neural recording of cardiac vagal control. Therefore, our conclusions still need to be correlated with and validated by more direct evaluation of the activity of the autonomic nervous system, such as peroneal microneurography or plasma catecholamines measurements. Future study protocols will need to verify the current data.

## 5. Conclusions

In conclusion, both IPC maneuvers with high and low pressure acutely increased the maximum strength in varied resistance exercises compared with the CON condition. Our findings showed that there seems to be a physiological effect acting through the sympathovagal balance evaluated through heart rate variability, which may explain, at least in part, the psycho-physiological hypothesis argued in previous studies [5,9,10]. This physiological effect may explain the increased performance in both experimental conditions when compared with the CON protocol. Therefore, a total occlusion may not be necessary to cause the acute effects on the maximum strength in every individual.

For coaches of strength and conditioning sports who seek new strategies to improve athlete’s competition performance and neuromuscular adaptations, we recommend the use of IPC before resistance training and before competitions with the objective of increasing maximum strength. In this study, the interventions consisted of 4 × 5 min of occlusion with two intensities (20 and 220 mmHg) alternated with 4 × 5 min of reperfusion, totaling 40 min of pre-conditioning, and both improved maximum strength. Coaches may choose to substantially decrease the occlusion pressure considering the athlete’s perception of discomfort.

## Figures and Tables

**Figure 1 ijerph-19-07655-f001:**
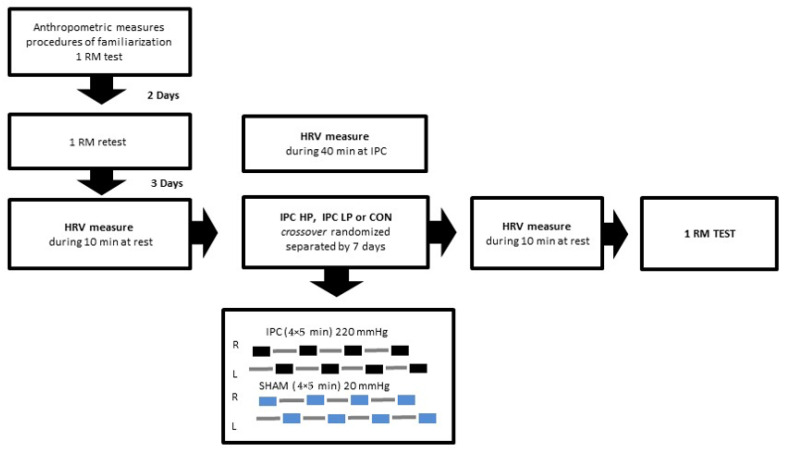
Study design. 1 RM TEST: one maximum repetition testing; IPC HP: ischemic preconditioning high pressure (220 mmHg); IPC LP: ischemic preconditioning low pressure (20 mmHg); CON: control protocol; HRV: heart rate variability.

**Figure 2 ijerph-19-07655-f002:**
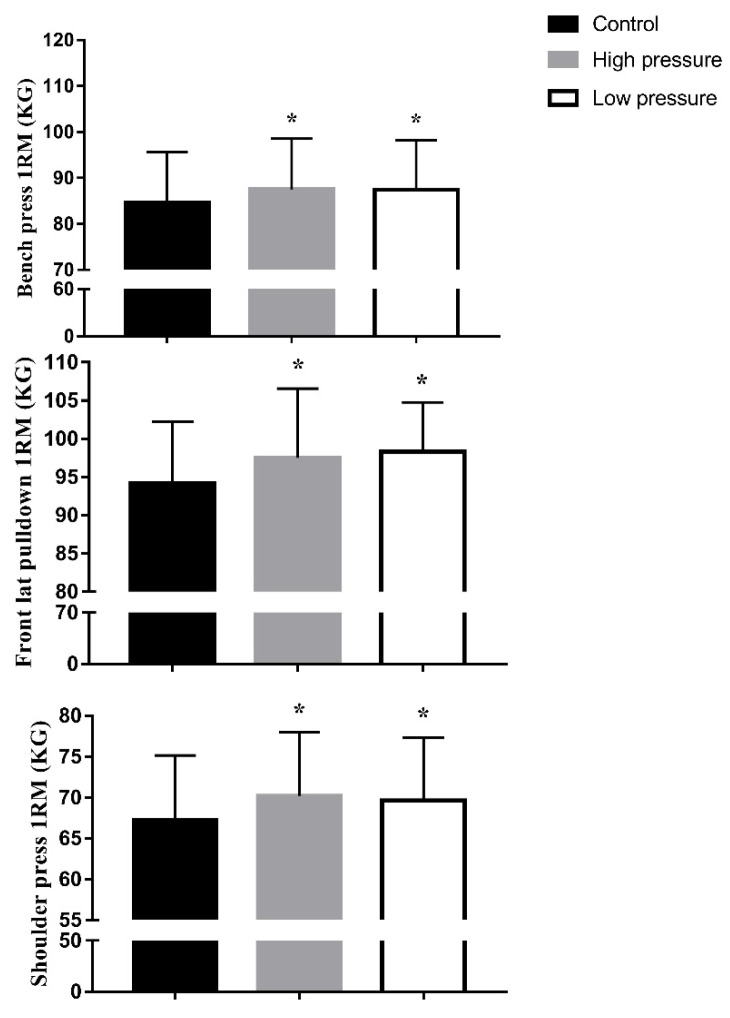
Maximal strength in control and IPC high and low pressure. 1 RM: one maximum repetition; high pressure: ischemic preconditioning high pressure (220 mmHg); low pressure: ischemic preconditioning low pressure (20 mmHg); * *p* < 0.05 vs. baseline.

**Figure 3 ijerph-19-07655-f003:**
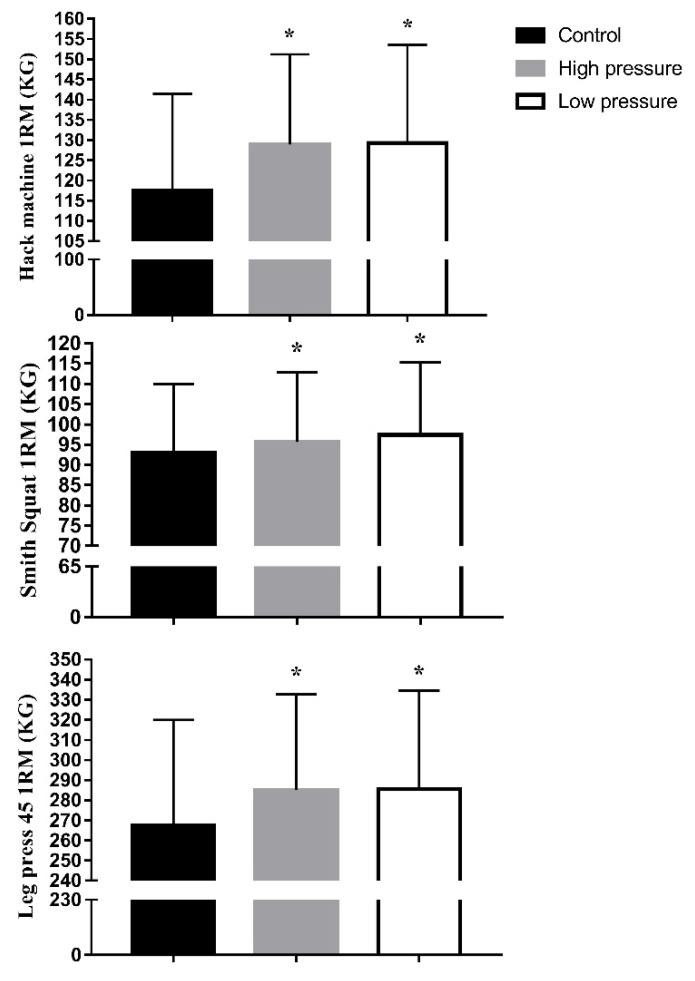
1 RM in control and IPC high and low pressure. 1 RM (one maximum repetition) in control and after 40 min if ischemic preconditioning with high (220 mmHg) and low (20 mmHg) pressure. * *p* < 0.05 vs. control.

**Figure 4 ijerph-19-07655-f004:**
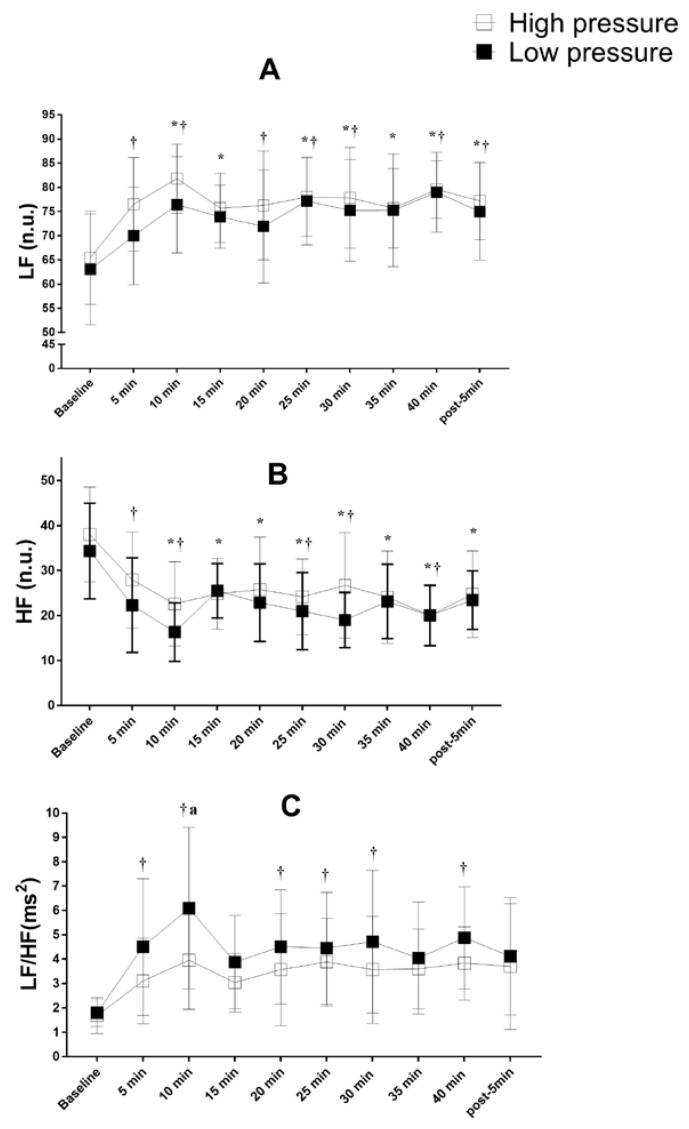
LF, HF, and LF/HF ratio during and after IPC high and low pressure. (**A**) LF = low frequency in normalized units; (**B**) HF = high frequency in normalized units; (**C**) LF/HF ratio; high pressure: ischemic preconditioning protocol 220 mmHg; low pressure: ischemic preconditioning protocol 20 mmHg; * *p* < 0.05 IPC_high_ vs. baseline; ^†^
*p* < 0.05 IPC_low_ vs. baseline; ^a^
*p* < 0.05 IPC_high_ vs. IPC_low_.

**Figure 5 ijerph-19-07655-f005:**
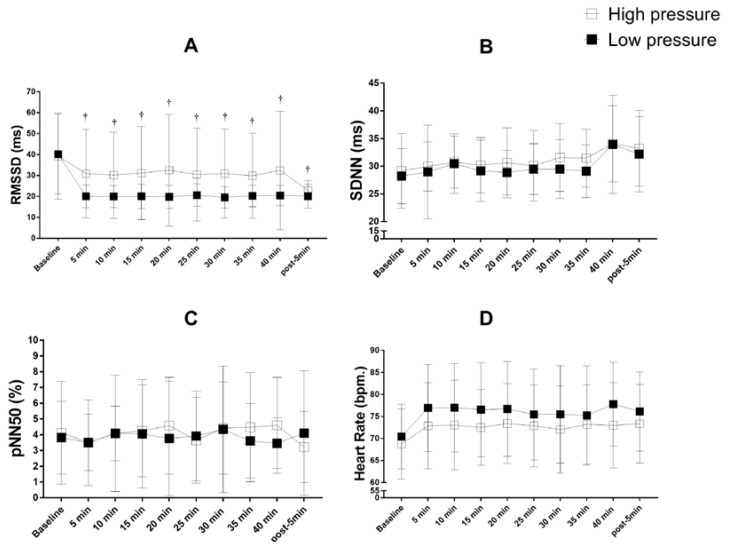
RMSSD and pND50 during and after IPC high and low pressure. (**A**) RMSSD; (**B**) SDNN; (**C**) pNN50; (**D**) heart rate; high pressure: ischemic preconditioning protocol 220 mmHg; low pressure: ischemic preconditioning protocol 20 mmHg; ^†^
*p* < 0.05 IPC_low_ vs. baseline.

**Table 1 ijerph-19-07655-t001:** Characteristics of subjects (*n* = 16).

Age (years)	27.8 ± 3.8
Height (cm)	180.4 ± 6.1
Weight (kg)	82.6 ± 8.8
BMI (kg m^−2^)	25.3 ± 1.8
Training history (years)	2.4 ± 1.0
Bench press 1 RM (kg)	84.6 ± 10.6
Leg Press 1 RM (kg)	267.4 ± 50.8
Lat pull down 1 RM (kg)	94.2 ± 7.8
Hack machine 1 RM (kg)	117.5 ± 23.2
Shoulder press 1 RM (kg)	70.2 ± 7.6
Back squat 1 RM (kg)	92.9 ± 16.5
LF_nu_	62.5 ± 9.2
HF_nu_	40.2 ± 10.1
LF/HF ratio	1.6 ± 0.6
RMSSD (ms)	41.9 ± 19.6
SDNN (ms)	30.5 ± 4.0
pNN50 (%)	4.4 ± 2.9
Heart rate (BPM)	66.1 ± 11.9

BMI: body mass index; 1 RM: one repetition maximum; LF = low frequency in normalized units; HF = high frequency in normalized units; LF/HF ratio; RMSSD (ms) = square root of the sum of the square of the differences between the R-R intervals divided by the number of R-R intervals; SDNN (standard deviation of all normal RR intervals (ms); pNN50 corresponds to the percentage difference between adjacent NN intervals that are greater than 50 ms.

**Table 2 ijerph-19-07655-t002:** IPC high and low pressure effects on HRV variables.

	IPC High Pressure				IPC Low Pressure		
	*P*	ES	Δ%	*P*	ES	Δ%	*P*
	LF-n.u.						
20 min During	0.88	0.80	14.09	0.19	1.16	16.55	0.04
5 min Post	1.00	1.08	18.92	0.01	1.26	17.98	0.03
	HF-n.u.						
20 min During	1.00	−1.24	−31.10	0.01	−1.13	−36.91	0.07
5 min Post	1.00	−1.03	−36.84	0.01	−0.56	−39.47	0.84
LF/HF							
20 min During	0.91	2.58	110.89	0.28	4.86	149.85	0.01
5 min Post	1.00	2.75	118.53	0.20	4.14	127.87	0.07
RMSSD ms.							
20 min During	0.25	−0.26	−17.22	0.98	−3.81	−50.62	0.02
5 min Post	1.00	−3.55	−41.61	0.13	−3.62	−49.83	0.02
SDNN ms.							
20 min During	1.00	0.24	5.08	1.00	0.16	2.26	1.00
5 min Post	1.00	0.62	14.04	0.65	0.60	14.09	1.00
pNN50 (%)							
20 min During	1.00	0.15	10.74	1.00	−0.02	−1.63	1.00
5 min Post	1.00	−0.42	−22.09	1.00	0.07	7.35	1.00
Heart rate (BPM)							
20 min During	1.00	0.60	6.72	0.94	0.88	8.91	0.71
5 min Post	1.00	0.60	6.67	0.94	0.80	8.06	0.82

IPC high pressure: ischemic preconditioning protocol 220 mmHg; IPC low pressure: ischemic preconditioning protocol 20 mmHg; ES = effect size; Δ% = difference between post and baseline moments in percentage; LF = low frequency in normalized units; HF = high frequency in normalized units; LF/HF ratio; RMSSD_ms_ = square root of the sum of the square of the differences between the R-R intervals divided by the number of R-R intervals; SDNN (standard deviation of all normal RR intervals (ms)); pNN50 corresponds to the percentage difference between adjacent NN intervals that are greater than 50 ms.

## Data Availability

Available through the corresponding author by request.

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
