# Peer review of "Ischemic Preconditioning with High and Low Pressure Enhances Maximum Strength and Modulates Heart Rate Variability"

_ijerph, 2022, doi:10.3390/ijerph19137655_

Round 1

Reviewer 1 Report

General Comments: Although the topic of this work is of interest, the study design appears to be flawed as the baseline trial was completed prior to the two subsequent trials (IPCHigh and IPCLow). With this in-mind, it is very likely that a learning effect could have occurred during the experiment that might  have influenced the reported increase in 1RM. Although changes in sympathovagal balance within the trail are predictable, the evidence presented does not appear to support a consistent difference in this pattern between trials. Therefore, the conclusion to this work appears very difficult to justify: “High and low pressures used during the IPC maneuver influence the autonomic nervous system, which partly explain the maximal strength enhancement”.

Specific Comments:

  1. Abstract (line 20-): the first sentence needs to provide a rationale for this work – this statement needs reworking.
  2. Abstract (line 22-): although I do not disagree with the statement that the study might be able to compare high and low pressure IPC, the majority of the results and discussion are focusing on comparison to control, which is likely flawed as stated in my general comments.
  3. Abstract (line 35-): this conclusion is not supported by data presented and needs reworking.
  4. Introduction (lines 78-85): doesn’t this paragraph suggest that data might exist to answer your question? If so, the introduction should focus on these data rather than the training effects and build a clearer rationale for the current study.
  5. Introduction (lines 86-95): While I support the use of HRV as an indirect estimate of sympathovagal balance (note spelling), it has been clearly demonstrated that the method cannot evaluate parasympathetic and sympathetic activity. With this in-mind, this paragraph needs tweaking to avoid confusion.
  6. Methods (experimental design): Why were the three trials (1RM, IPC HP and IPC LP) not completed in a random order? As previously described, this is a major concern when interpreting the results that are presented in this manuscript. Also, the naming of these trials is inconsistent in the text and figure 1.
  7. Methods and results: naming the first condition (no IPC) as baseline is problematic in this acute study. You are not evaluating changes over time; in fact, you should be trying to reduce the influence of time on this study design.
  8. Methods (2.2): Please provide further clarity about the model that was used in the sample size calculation. It would be useful also to use some of the previous studies mentioned in the introduction to estimate the effect size anticipated using appropriate data to support the stated effect size.
  9. Methods (2.3.4): I suggest that you reword the first sentence. I assume that you continuously recorded and then sectioned these data. How did you control the environment and participant conditions during this experiment to maintain constant covariates. More detail is necessary to explain removal of noise - this is very important in short-term HRV recordings. I suggest that you re-word the sections that describe alignment of measures in the frequency domain with sympathetic and parasympathetic components as this is problematic.
  10. Results (line 226-228): I assume that these ICC values represent familiarisation to baseline 1 RM data. ICC are not particularly useful in this instance, I would be more interested in seeing measurements of the absolute reliability.
  11. Table 2: this table is very messy. The attempt to include Baseline measures in the same table as change has meant that the column headings do not match data presented. Additionally, the table title is not descriptive of these data.
  12. Results: in my opinion, differences from what is currently called baseline are too difficult to interpret due to the potential order effect. As the majority of the findings appear to focus on these, I am not convinced that the results section is useful in its current form.
  13. Discussion: I have not reviewed the discussion because I am not confident that the results can be interpreted as they are stated currently.
  14. Conclusions: I am not confident that the current conclusions reflect the findings of this study.

Author Response

Dear Reviewer

We are grateful for your consideration of this manuscript, and we also very much appreciate your suggestions, which have been very helpful in improving the manuscript. We also thank the reviewers for their careful reading of our text. All the comments we received on this study of all reviewers have been attended into account in improving the quality of the article, and we present our reply to each of them separately.

General Comments: Although the topic of this work is of interest, the study design appears to be flawed as the baseline trial was completed prior to the two subsequent trials (IPCHigh and IPCLow). With this in-mind, it is very likely that a learning effect could have occurred during the experiment that might have influenced the reported increase in 1RM. Although changes in sympathovagal balance within the trail are predictable, the evidence presented does not appear to support a consistent difference in this pattern between trials. Therefore, the conclusion to this work appears very difficult to justify: “High and low pressures used during the IPC maneuver influence the autonomic nervous system, which partly explain the maximal strength enhancement”.

Specific Comments:

Abstract (line 20-): the first sentence needs to provide a rationale for this work – this statement needs reworking.

A: We add sentence.

Abstract (line 22-): although I do not disagree with the statement that the study might be able to compare high and low pressure IPC, the majority of the results and discussion are focusing on comparison to control, which is likely flawed as stated in my general comments.

Abstract (line 35-): this conclusion is not supported by data presented and needs reworking. we can rewrite like this: In conclusion, both IPC maneuvers increased the maximum strength in varied resistance exercises in an acute manner compared to the control values.

A: Our wording was not clear enough and may lead to a different interpretation. Study design section has been rewritten.

“The present study was performed in 5 visits to the laboratory, separated by 3 to 7 days, always at the same time of day to avoid the circadian influence (figure 1). In the first visit all participants signed the informed consent form and fulfilled the Physical Activity Readiness Questionnaire / PAR-Q. Anthropometric measures and familiarization with 1 repetition maximum (1RM) testing procedures were also performed in the first visit. During the second visit 1RM retest was performed. In the third, fourth and fifth visit all participants were randomly assigned to three experimental protocols: a) high pressure IPC (220 mmHg) (IPChigh) + 1 RM testing; b) low pressure IPC protocol (20 mmHg) (IPClow) + 1 RM testing; c) 1RM (control protocol). “

Introduction (lines 78-85): doesn’t this paragraph suggest that data might exist to answer your question? If so, the introduction should focus on these data rather than the training effects and build a clearer rationale for the current study.

A: Yes, there is data to answer the question. The introduction focuses on these data, but also attempts to justify the connection of HRV to the effects of low-pressure IPC on maximum force.

Introduction (lines 86-95): While I support the use of HRV as an indirect estimate of sympathovagal balance (note spelling), it has been clearly demonstrated that the method cannot evaluate parasympathetic and sympathetic activity. With this in-mind, this paragraph needs tweaking to avoid confusion.

A: We rewrite. “Heart rate variability has been used to indirectly and non-invasively estimate of sympathovagal balance after application of IPC [20, 21, 22].”

Methods (experimental design): Why were the three trials (1RM, IPC HP and IPC LP) not completed in a random order? As previously described, this is a major concern when interpreting the results that are presented in this manuscript. Also, the naming of these trials is inconsistent in the text and figure 1.

A: The study was randomized.

Methods and results: naming the first condition (no IPC) as baseline is problematic in this acute study. You are not evaluating changes over time; in fact, you should be trying to reduce the influence of time on this study design.

A: Our wording was not clear enough and may lead to a different interpretation. Study design section has been rewritten.

“The present study was performed in 5 visits to the laboratory, separated by 3 to 7 days, always at the same time of day to avoid the circadian influence (figure 1). In the first visit all participants signed the informed consent form and fulfilled the Physical Activity Readiness Questionnaire / PAR-Q. Anthropometric measures and familiarization with 1 repetition maximum (1RM) testing procedures were also performed in the first visit. During the second visit 1RM retest was performed. In the third, fourth and fifth visit all participants were randomly assigned to three experimental protocols: a) high pressure IPC (220 mmHg) (IPChigh) + 1 RM testing; b) low pressure IPC protocol (20 mmHg) (IPClow) + 1 RM testing; c) 1RM (control protocol). “

Methods (2.2): Please provide further clarity about the model that was used in the sample size calculation. It would be useful also to use some of the previous studies mentioned in the introduction to estimate the effect size anticipated using appropriate data to support the stated effect size.

A: We rewrite.

Methods (2.3.4): I suggest that you reword the first sentence. I assume that you continuously recorded and then sectioned these data. How did you control the environment and participant conditions during this experiment to maintain constant covariates. More detail is necessary to explain removal of noise - this is very important in short-term HRV recordings. I suggest that you re-word the sections that describe alignment of measures in the frequency domain with sympathetic and parasympathetic components as this is problematic.

A: We rewrite this section.

Results (line 226-228): I assume that these ICC values represent familiarisation to baseline 1 RM data. ICC are not particularly useful in this instance, I would be more interested in seeing measurements of the absolute reliability.

A: The ICC values are from all 1RM test and retest.

Absolute reliabitlity:

Bench press: Test 83.6±11.1kg and retest 84.6±10.6 kg

Leg press 45: test 262.5±47.5 kg and retest 267.4±50.8 kg

Front Lat pulldown: test 92.2±8.0kg and retest 94.2±7.8 kg

Hack machine: test 114.9±22.8 kg and retest 117.5±23.2 kg

Shoulder press test 66.8±7.7kg and retest 67.2±7.7kg

Smith Squat test 90.3±17.2kg and retest 92.9±16.5 kg

Table 2: this table is very messy. The attempt to include Baseline measures in the same table as change has meant that the column headings do not match data presented. Additionally, the table title is not descriptive of these data.

A: We improved Table 2 and change the title to “IPC High and Low pressure effects on HRV variables.”

Results: in my opinion, differences from what is currently called baseline are too difficult to interpret due to the potential order effect. As the majority of the findings appear to focus on these, I am not convinced that the results section is useful in its current form.

Discussion: I have not reviewed the discussion because I am not confident that the results can be interpreted as they are stated currently.

Conclusions: I am not confident that the current conclusions reflect the findings of this study.

A: Our wording was not clear enough and may lead to a different interpretation. Study design section has been rewritten and now we think with the changes made we improved the article.

Reviewer 2 Report

The original article titled ‘Ischemic preconditioning with high and low pressure enhances 2 maximum strength and modulates heart rate variability’ reports acute effects of ischemic preconditioning before resistance training in improving the maximal strength and also shows modulation of heart rate variability by both high- and low-pressure IPC. The paper also indicates achieving significant benefits with low pressure comparable to total occlusion at least in context of acute strategies. The article is well written, the aims and objectives are clearly stated and the results and discussion sections are built around the objective of the study. I have few queries and suggestions as below:

  1. Since the entire study is focused on IPC effects on resistance training/exercises, I think the title of the article should also include this category.

  1. Did the authors collect the information regarding cardiovascular or metabolic disease for all participants? It should be a part of exclusion criteria as well.

  1. By looking at the study design it seems that the authors control data is based on baseline readings taken on visit 2, while the experimental data came from two subsequent visits: visit 3 and 4. How do the authors justify not having the baseline or control data from the same day when experimental group was tested. The authors should clarify this better in the text or discuss the possible limitations.

  1. The study design in Figure 1 is represented well. I however feel it could be improved by indicating the four visits in representation as well. And though all participants were available for each visit, it would still be important to show the number of participants at each visit by indicating n=16. Also indicate at what point baseline data was collected (visit 2).

  1. Table 2 is not well represented. In the pdf format of the file, with respect to first column, I am unable to discern between the subsequent rows. It could be a formatting issue, but please ensure that the information is legible and readable. I was unable to follow the information in table 2.

  1. Please maintain conformity across text and figures/tables in writing IPClow or IPClow. Same for IPChigh or IPChigh

  1. There are some errors in text with respect to spellings or repetitions. Please check the text and figures thoroughly to avoid these.

I believe that having addressed my queries and suggested revision, the paper would be a good addition to the field addressing the acute effects of Ischemic preconditioning.

Author Response

Dear Reviewer

We are grateful for your consideration of this manuscript, and we also very much appreciate your suggestions, which have been very helpful in improving the manuscript. We also thank the reviewers for their careful reading of our text. All the comments we received on this study of all reviewers have been attended into account in improving the quality of the article, and we present our reply to each of them separately.

The original article titled ‘Ischemic preconditioning with high and low pressure enhances 2 maximum strength and modulates heart rate variability’ reports acute effects of ischemic preconditioning before resistance training in improving the maximal strength and also shows modulation of heart rate variability by both high- and low-pressure IPC. The paper also indicates achieving significant benefits with low pressure comparable to total occlusion at least in context of acute strategies. The article is well written, the aims and objectives are clearly stated and the results and discussion sections are built around the objective of the study. I have few queries and suggestions as below:

Since the entire study is focused on IPC effects on resistance training/exercises, I think the title of the article should also include this category

A: Thank you for the suggestion. The maximal strength test already implies that the study is focused on resistance exercise. Therefore, in order not to make the title too long, we prefer to keep the current title.

Did the authors collect the information regarding cardiovascular or metabolic disease for all participants? It should be a part of exclusion criteria as well.

A: We did not collect but we asked. Although with the application of PAR-Q questionnaire we exclude any possible major risk participant.

By looking at the study design it seems that the authors control data is based on baseline readings taken on visit 2, while the experimental data came from two subsequent visits: visit 3 and 4. How do the authors justify not having the baseline or control data from the same day when experimental group was tested. The authors should clarify this better in the text or discuss the possible limitations.

A: Our wording was not clear enough and this section has been rewritten.

“The present study was performed in 5 visits to the laboratory, separated by 3 to 7 days, always at the same time of day to avoid the circadian influence (figure 1). In the first visit all participants signed the informed consent form and fulfilled the Physical Activity Readiness Questionnaire / PAR-Q. Anthropometric measures and familiarization with 1 repetition maximum (1RM) testing procedures were also performed in the first visit. During the second visit 1RM retest was performed. In the third, fourth and fifth visit all participants were randomly assigned to three experimental protocols: a) high pressure IPC (220 mmHg) (IPChigh) + 1 RM testing; b) low pressure IPC protocol (20 mmHg) (IPClow) + 1 RM testing; c) 1RM (control protocol). “

The study design in Figure 1 is represented well. I however feel it could be improved by indicating the four visits in representation as well. And though all participants were available for each visit, it would still be important to show the number of participants at each visit by indicating n=16. Also indicate at what point baseline data was collected (visit 2).

A: We changed the figure 1.

Table 2 is not well represented. In the pdf format of the file, with respect to first column, I am unable to discern between the subsequent rows. It could be a formatting issue, but please ensure that the information is legible and readable. I was unable to follow the information in table 2.

 A: We formatted the table 2.

Please maintain conformity across text and figures/tables in writing IPClow or IPClow. Same for IPChigh or IPChigh

A: We changed

There are some errors in text with respect to spellings or repetitions. Please check the text and figures thoroughly to avoid these.

 A: We done.

I believe that having addressed my queries and suggested revision, the paper would be a good addition to the field addressing the acute effects of Ischemic preconditioning.

Reviewer 3 Report

General Comment: Good study investigating the effect of ischemic preconditioning. Very nice approach to evaluate its influence on the autonomic nervous system by measuring the HRV.

Specific Comment:

Line 103: Please report the average day of separation

Line 104: Please report the detail of ethical approval

Line 218: Should it be “Cohen's d”.

Line 298-300: No sure the actual meaning of “enhance muscle performance more than a placebo application with low pressure”. Please explain and rephrase.

Discussion: Great if more elaboration can be done on the application of sports coaching.

Author Response

Dear Reviewer

We are grateful for your consideration of this manuscript, and we also very much appreciate your suggestions, which have been very helpful in improving the manuscript. We also thank the reviewers for their careful reading of our text. All the comments we received on this study of all reviewers have been attended into account in improving the quality of the article, and we present our reply to each of them separately.

General Comment: Good study investigating the effect of ischemic preconditioning. Very nice approach to evaluate its influence on the autonomic nervous system by measuring the HRV.

Specific Comment:

Line 103: Please report the average day of separation

A: We add.

Line 104: Please report the detail of ethical approval

A: This “umbrella” project involves several subprojects (as listed in the specific objectives). The study will be of the “association with interference” type, according to Volpato (2011), because we will analyze possible interference between independent variables (specific training/test/protocol and type of strategy: ischemic preconditioning applied at rest, before exercises, about dependent variables. Number: 90060318.0.0000.5237

Line 218: Should it be “Cohen's d”.

A: We changed.

Line 298-300: No sure the actual meaning of “enhance muscle performance more than a placebo application with low pressure”. Please explain and rephrase.

A: We rewrite. “Although high pressures induce local hypoxia, benefiting from the hypoxia- and reperfusion-related complex cascade event, not always promote higher muscle performance than a placebo application with low pressure”

Discussion: Great if more elaboration can be done on the application of sports coaching.

A: We add practical applications at the end of conclusions.

Round 2

Reviewer 1 Report

General Comments: Although the authors have made some changes to the introduction and results section to improve readability, the findings and discussion focus on comparison of the two conditions (IPCHigh and IPCLow) against an initial condition (now referred to as control). As previously stated, this design is flawed as the control trial was completed prior to the two subsequent trials. Although changes in sympathovagal balance within the trial are predictable, the evidence presented does not appear to support a consistent difference in this pattern between trials. Therefore, the conclusion to this work appears very difficult to justify: “High and low pressures used during the IPC maneuver influence the autonomic nervous system, which partly explain the maximal strength enhancement”. The authors have not carried forward there acknowledgement of my concerns in the interpretation of data and HRV analyses into the discussion, which has many instances of misleading interpretation.

Specific Comments:

1. Abstract (line 20-): the first sentence needs to provide a rationale for this work – this statement needs reworking.

2. Abstract (line 22-): this sentence needs focusing – the present sentence is too vague to be of use to develop the rationale.

3. Abstract (line 35-): this conclusion is not supported by data presented and needs reworking.

4. Introduction (lines 78-85): The introduction should focus on these data rather than the training effects and build a clearer rationale for the current study.

5. Introduction (lines 86-95): While my suggested adjustment to wording has been implemented here, the discussion (see introductory paragraph carries forward the misconception that HRV can provide a measurement of sympathetic and parasympathetic activities – this has been repeatedly demonstrated in the literature to be impossible.

6. Methods (experimental design): Why were the three trials (1RM, IPC HP and IPC LP) not completed in a random order? As previously described, this is a major concern when interpreting the results that are presented in this manuscript. Although the naming of these trials has been made more consistent, as previously described, it is very problematic to use the first trial as a control.

8. Methods (2.2): Please provide further clarity about the model that was used in the sample size calculation. The revised manuscript information on sample size calculation does not present the model used or the justification of an anticipated effect size of 0.32. Therefore, it is unclear if the study is sufficiently powered to determine a difference between IPC HP and IPC LP.

10. Results (previously lines 226-228): The authors have provided means and SD values but without seeing measurements of the absolute reliability it is difficult to determine what SMD is possible in this work.

12. Results: in my opinion, differences from what is now called control are still too difficult to interpret due to the potential order effect. As the majority of the findings appear to focus on these, I am still not convinced that the results section is useful in its current form.

12. Discussion: It is disappointing that the authors have not revised the discussion based on concerns raised in the previous review. This is important because not addressing these concerns in the discussion has potential to lead readers to misinterpret the implications of the findings.

13. Discussion: all references to differences from ‘control’ condition are not valid as the design does not enable comparison between control and the other two interventions (IPC HP and IPC LP). In my view, the discussion should focus on differences between intervention conditions without reference to the prior trial.

14. Discussion:  As previously stated, the misconception that HRV can provide a measurement of sympathetic and parasympathetic activities must be avoided – this has been repeatedly demonstrated in the literature to be problematic. The authors have not made changes to the discussion to avoid propagating this misconception (e.g., opening paragraph of discussion and in lines 396-397).

18. Conclusions: Again, the conclusions are based on differences between two interventions (IPC HP and IPC LP) with the ‘control’. This is flawed and therefore I am not confident that the current conclusions reflect the findings of this study.

19. The document contains a number of errors in syntax and spelling that need addressing.

Author Response

Dear Reviewer,

We are grateful for your consideration of this manuscript, and we also very much appreciate your suggestions, which have been very helpful in improving the manuscript.

 General Comments: Although the authors have made some changes to the introduction and results section to improve readability, the findings and discussion focus on comparison of the two conditions (IPCHigh and IPCLow) against an initial condition (now referred to as control). As previously stated, this design is flawed as the control trial was completed prior to the two subsequent trials. Although changes in sympathovagal balance within the trial are predictable, the evidence presented does not appear to support a consistent difference in this pattern between trials. Therefore, the conclusion to this work appears very difficult to justify: “High and low pressures used during the IPC maneuver influence the autonomic nervous system, which partly explain the maximal strength enhancement”. The authors have not carried forward there acknowledgement of my concerns in the interpretation of data and HRV analyses into the discussion, which has many instances of misleading interpretation.

A: We respectfully believe the Reviewer was mistaken by confusion in the manuscript. The participants actually performed three protocols that were administered in random order: control, and IPC with high and low pressure. These protocols were separated by 7 days (please refer to figure 1). According to your comment and concern, we have clarified the abstract and the manuscript in several places to avoid confusion. This design is robust and allows us to assess the impact of pressure vs control.

Specific Comments:

  1. Abstract (line 20-): the first sentence needs to provide a rationale for this work – this statement needs reworking.

A: Yes, this has been added.

  1. Abstract (line 22-): this sentence needs focusing – the present sentence is too vague to be of use to develop the rationale.

A: We have reformulated this sentence.

  1. Abstract (line 35-): this conclusion is not supported by data presented and needs reworking.

A: Yes, this has been reworded.

  1. Introduction (lines 78-85): The introduction should focus on these data rather than the training effects and build a clearer rationale for the current study.

A: Yes, we have followed this reasoning and modified the introduction.

  1. Introduction (lines 86-95): While my suggested adjustment to wording has been implemented here, the discussion (see introductory paragraph carries forward the misconception that HRV can provide a measurement of sympathetic and parasympathetic activities – this has been repeatedly demonstrated in the literature to be impossible.

A: we agree with your suggestion, we have adjusted the discussion.

  1. Methods (experimental design): Why were the three trials (1RM, IPC HP and IPC LP) not completed in a random order? As previously described, this is a major concern when interpreting the results that are presented in this manuscript. Although the naming of these trials has been made more consistent, as previously described, it is very problematic to use the first trial as a control.

A: As mentioned above, we believe the Reviewer was mistaken due to confusion in the manuscript. The study was randomized into 3 interventions to avoid the effect of the order. This has been clarified.

  1. Methods (2.2): Please provide further clarity about the model that was used in the sample size calculation. The revised manuscript information on sample size calculation does not present the model used or the justification of an anticipated effect size of 0.32. Therefore, it is unclear if the study is sufficiently powered to determine a difference between IPC HP and IPC LP.

A: Agreed, this has been added.

  1. Results (previously lines 226-228): The authors have provided means and SD values but without seeing measurements of the absolute reliability it is difficult to determine what SMD is possible in this work.

A: The reliability values are indicated in the results section (in figures 4 and 5).

  1. Results: in my opinion, differences from what is now called control are still too difficult to interpret due to the potential order effect. As the majority of the findings appear to focus on these, I am still not convinced that the results section is useful in its current form.

A: This has been clarified. The study was randomized into 3 interventions to precisely avoid the effect of the order.

  1. Discussion: It is disappointing that the authors have not revised the discussion based on concerns raised in the previous review. This is important because not addressing these concerns in the discussion has potential to lead readers to misinterpret the implications of the findings.

A: The discussion has been modified according to the previous suggestions.

  1. Discussion: all references to differences from ‘control’ condition are not valid as the design does not enable comparison between control and the other two interventions (IPC HP and IPC LP). In my view, the discussion should focus on differences between intervention conditions without reference to the prior trial.

A: The control condition was randomized. We have rewritten this section.

  1. Discussion: As previously stated, the misconception that HRV can provide a measurement of sympathetic and parasympathetic activities must be avoided – this has been repeatedly demonstrated in the literature to be problematic. The authors have not made changes to the discussion to avoid propagating this misconception (e.g., opening paragraph of discussion and in lines 396-397).

A: Agreed, we have changed the discussion to avoid propagating this misconception

  1. Conclusions: Again, the conclusions are based on differences between two interventions (IPC HP and IPC LP) with the ‘control’. This is flawed and therefore I am not confident that the current conclusions reflect the findings of this study.

A: The design had 3 randomised interventions. Nonetheless, we slightly amended this section.

  1. The document contains a number of errors in syntax and spelling that need addressing.

A: We have addressed this in the revised version of the manuscript.